# Cross-Space Adaptive Filter: Integrating Graph Topology and Node Attributes for Alleviating the Over-smoothing Problem

## ABSTRACT

The vanilla Graph Convolutional Network (GCN) uses a low-pass filter to extract low-frequency signals from graph topology, which may lead to the over-smoothing problem when GCN goes deep. To this end, various methods have been proposed to create an adaptive filter by incorporating an extra filter (e.g., a high-pass filter) extracted from the graph topology. However, these methods heavily rely on topological information and ignore the node attribute space, which severely sacrifices the expressive power of the deep GCNs, especially when dealing with disassortative graphs. In this paper, we propose a cross-space adaptive filter, called CSF, to produce the adaptive-frequency information extracted from both the topology and attribute spaces. Specifically, we first derive a tailored attribute-based high-pass filter that can be interpreted theoretically as a minimizer for semi-supervised kernel ridge regression. Then, we cast the topology-based low-pass filter as a Mercer's kernel within the context of GCNs. This serves as a foundation for combining it with the attribute-based filter to capture the adaptive-frequency information. Finally, we derive the cross-space filter via an effective multiple-kernel learning strategy, which unifies the attribute-based high-pass filter and the topology-based low-pass filter. This helps to address the over-smoothing problem while maintaining effectiveness. Extensive experiments demonstrate that CSF not only successfully alleviates the over-smoothing problem but also promotes the effectiveness of the node classification task.

## CCS CONCEPTS

• **Computing methodologies** → *Neural networks*.

## KEYWORDS

Graph convolutional network, over-smoothing, node attribute

**ACM Reference Format:**
Anonymous Author(s). 2018. Cross-Space Adaptive Filter: Integrating Graph Topology and Node Attributes for Alleviating the Over-smoothing Problem. In *Proceedings of Make sure to enter the correct conference title from your rights confirmation emai (Conference acronym 'XX)*. ACM, New York, NY, USA, 13 pages. https://doi.org/XXXXXXX.XXXXXXX

**Relevance to The Web Conference**. This work studies the issue of over-smoothing in GNNs. As a result, it meets the requirements of *Graph Algorithms and Modeling for the Web* Track and is closely related to the topic of "*Graph embeddings and GNNs for the Web*".

## 1 INTRODUCTION

Graph Neural Networks (GNNs) have proven to be effective in learning representations of network-structured data and have achieved great success in various real-world web applications, such as citation networks [21, 34] and actor co-occurrence network [38]. As one of the mainstream research lines of GNNs, spectral-based methods have attracted much attention due to their strong mathematical foundation. Typically, these methods build upon graph signal processing and define the convolution operation using the graph Fourier transform. Recent studies suggest that superior performance can be achieved when jointly characterizing graph topology and node attributes [45, 47]. This is because node attributes contain rich information, such as the correlation among node attributes, which complements the information from graph topology.

As the seminal work of spectral-based GNNs, the Graph Convolutional Network (GCN) has been widely explored. Informally, GCN is a multi-layer feed-forward neural network that propagates node representations across an undirected graph. During the convolution operation, each node updates its representation by aggregating representations from its connected neighborhood. Although the effectiveness of GCN, it, unfortunately, suffers from the **over-smoothing problem** [32, 50], where node representations become indistinguishable and converge towards the same constant value as the number of layers increases. This is because the convolution operation in a GCN layer is governed by a low-pass spectral filter, which causes connected nodes in the graph to share similar representations [1, 42]. Previous studies show that this low-pass filter corresponds to the eigensystem of the graph Laplacian and penalizes large eigenvalues in its eigen-expansion. This helps to remove un-smooth signals from the graph and ensures that connected nodes tend to share similar representations [35]. However, as the number of layers increases, the over-smoothing problem occurs. To address this issue, researchers have been striving to move beyond the low-pass filter and create an adaptive spectral filter [6]. This is achieved by learning an additional matrix-valued function on the eigenvalues of the graph Laplacian that produces an all-pass [19] or high-pass filter [2, 10]. These new filters are then integrated with the low-pass filter to yield the adaptive one. By this means, the adaptive filter incorporates adaptive frequency information rather than relying exclusively on low-pass frequency information.

However, existing adaptive filters only focus on the graph topology space but largely ignore the node attribute space, which severely sacrifices the expressive power of the deep GCNs, especially when the node's label is primarily determined by its own attributes rather than the topology. For example, when it comes to proteins, different types of amino acids often interact chemically with each other. Similarly, in actor co-occurrence networks, actor collaboration often occurs among different types of actors. In this case, instead of relying solely on the graph topology to determine which nodes should not have similar representations, considering the

correlation between node attributes can provide valuable prior knowledge on the dissimilarity between nodes. When ignoring the node attributes, previous studies show that the correlations of the learned node representations are potentially inconsistent with those of the raw node attributes [46, 48], In this case, the original node attributes are washed away, which leads to decreased performance [18]. This is particularly true when dealing with disassortative graphs [51] where neighboring nodes have dissimilar attributes or labels. Therefore, it is important to integrate both graph topology and node attribute spaces to alleviate the over-smoothing problem while at the same time promoting the effectiveness of downstream tasks. However, despite the recognition of its importance [32], further research efforts are required to fully realize it.

To this end, we propose a novel Cross-Space Filter (CSF, for short), an adaptive filter that integrates the adaptive-frequency information across both topology and node attribute spaces. In addition to the conventional low-pass filter extracted from the graph topology space, we first leverage the correlations of node attributes to extract a high-pass filter. Unlike other high-pass filters that are usually designed arbitrarily without model interpretation [2, 19], our high-pass filter, arising from a Mercer's kernel, is interpreted as a minimizer for semi-supervised kernel ridge regression. This brings more model transparency for humans to understand what knowledge the GCN extracts to make the specific filter. Then, to tackle the challenge of merging information from two separate spaces, we resort to the graph kernel theory [36] to cast the conventional low-pass filter of GCN into a kernel, unifying the two filters in Reproducing Kernel Hilbert Space (RKHS). This allows us to utilize the benefits of two spaces simultaneously. Subsequently, the proposed adaptive filter CSF is obtained by applying a simple multiple-kernel learning technique to fuse the information in both the topology and attribute spaces. As such, we successfully take advantage of both graph topology and node attribute spaces from the perspective of Mercer's kernel. Consequently, our CSF alleviates the over-smoothing problem while at the same time promoting the effectiveness of deep GCNs, especially on disassortative graphs. This provides insight into revisiting the role of node attributes and kernels in alleviating the over-smoothing problem.

To evaluate the effectiveness of CSF, we conduct comparative experiments with various baselines under different numbers of convolution layers. These experiments are conducted on both assortative and disassortative graphs to verify the superiority of CSF on different types of graphs. Compared to baselines, the results demonstrate that CSF not only successfully alleviates the over-smoothing problem by extracting and integrating information from both spaces but also improves the model performance on the downstream classification tasks, especially when dealing with disassortative graphs. In particular, on average, our CSF outperforms the best baseline by +0.62 on assortative graphs and +10.39 on disassortative graphs. These results highlight the importance of node attributes in assisting over-smoothing alleviation while at the same time promoting the effectiveness of deep GCNs. To sum up, we claim the following contributions.

- We call attention to the importance of node attribute space, which helps alleviate the over-smoothing problem while at the same time promoting the effectiveness of deep GCN, especially on disassortative graphs.

- For the first time, we leverage the correlations of node attributes to extract a spectral high-pass filter, arising from a Mercer's kernel. Such a filter could be further interpreted as a minimizer of semi-supervised kernel ridge regression, which brings more model transparency.

- We take the first step to derive a cross-space adaptive filter, which integrates the adaptive-frequency information across both topology and node attribute spaces. This provides insight into revisiting the role of node attributes and kernels in alleviating the over-smoothing problem.

- Extensive experiments on various datasets indicate that our method outperforms others in terms of its robustness to the over-smoothing problem and effectiveness on the downstream tasks, especially on the disassortative graphs.

## 2 RELATED WORK

**Graph Filters and GCN.** The fitted values/representations from GCN and its variants are reduced by a low-pass filter, corresponding to the eigensystem of graph Laplacian [1, 42], which results in the over-smoothing problem [50]. As a result, various adaptive filter-based methods have been proposed to extend the low-pass filter in GCN. Specifically, they focus on the combinations of multiple low-pass filters [12, 43], all-pass and low-pass filters [19], low-pass and band-pass filters [28, 51], and low-pass and high-pass filters [2, 10, 50]. In this paper, our method also combines low-pass and high-pass filters to eliminate the over-smoothing problem, but it differs from existing methods. For example, PGNN [10] designs a new propagation rule based on $p$-Laplacian message passing that works as low-high-pass filters. More recently, FAGCN [2] proposes a self-gating mechanism to achieve dynamic adaptation between low-pass and high-pass filters. Though considered in the model, the high-pass filter is based on a hand-crafted function, which is designed too arbitrarily without any interpretation. More importantly, existing works place heavy reliance on the graph topology to derive their adaptive filters, while the correlation information contained in the node's attributes is largely ignored. This may severely sacrifice the expressive power of the deep GCNs. This is particularly true when dealing with the disassortative graphs, where the node's label is primarily determined by its own attributes rather than the topology. To this end, we take the first step to derive a cross-space adaptive filter, which integrates information from both the topology and attribute spaces.

**Attribute-enhanced GCN.** Recent work shows that, together with the node attributes, superior performance can be achieved by characterizing graph topology and attribute correlations simultaneously [45, 47], due to node attributes containing abundant information that complements the graph topology. The correlations could be built using the predict-then-propagate architecture [13] or mutual exclusion constraints [46]. However, to the best of our knowledge, little research has attempted to use node attributes to address the over-smoothing problem. Although there have been some suggestions of adding residual connections to deep GNNs [32], such as the jumping knowledge [44], that is helpful for the over-smoothness problem, we differ from these methods in completely different technical solutions: we leverage the correlations of node attributes to extract a spectral high-pass filter for the first, and we experimentally show our superiority.

## 3 PRELIMINARY

**Notation.** Consider an undirected graph $G = (V, E, X)$ with adjacency matrix $A$, edge set $E$, node set $V$ with $|V| = N$. Also, the graph $G$ contains a node attribute matrix $X \in R^{N \times M}$, where each node $v_i \in V$ has an attribute vector $x_i \in R^M$. Moreover, we denote $\tilde{A} = A + I$ to be the adjacency matrix of graph $G$ with additional self-connections. $\tilde{D}$ and $D$ are defined as the diagonal degree matrix of $\tilde{A}$ and $A$ respectively, with $\tilde{D}_{ii} = \sum_j \tilde{A}_{ij}$ and $D_{ii} = \sum_j A_{ij}$. $L$ and $\tilde{L}$ are the normalized Laplacian matrix of $A$ and $\tilde{A}$, respectively. Also, unless stated otherwise, we denote $\{\lambda_i, v_i\}$ and $\{\tilde{\lambda}_i, \tilde{v}_i\}$ to be the $i$-th eigenvalue and eigenvector of $L$ and $\tilde{L}$, respectively. Next, we use the notation $[A, B]$ to be the concatenation operator between matrices or vectors. Finally, we assume $\mathbb{H}$ is a Reproducing Kernel Hilbert Space (RKHS) with a positive definite kernel function implementing the inner product. The inner product is defined so that it satisfies the reproducing property.

**Graph Convolutional Network.** A GCN is a multi-layer feedforward neural network that propagates and transforms node attributes along with an undirected graph $G$. The layer-wise propagation rule in layer $k$ is $H^{k+1} = \sigma(\tilde{D}^{-\frac{1}{2}} \tilde{A} \tilde{D}^{-\frac{1}{2}} H^k W^k)$. Here, $W^k$ is the trainable model parameter in layer $k$, $\sigma$ is an activation function (e.g., ReLU), $H^k$ is the hidden representations in the $k$-th layer, and $H^0 = X$ for initialization. To stabilize the optimization, the GCN adds self-loops to each node to make the largest eigenvalue of normalized Laplacian smaller [42]. Research has demonstrated that vectors containing fitted values of the GCN and its variants can be reduced by a customized low-pass filter, corresponding to the eigensystem of $\tilde{L}$. Specifically, Wu et al. [42] concludes that the low-pass filter for the simplified GCN is parameterized by the matrix-valued filter function $g(\tilde{\lambda}_i) = (1 - \tilde{\lambda}_i)^c$, where $c$ is the number of graph convolution layers. More accurately, the matrix-valued filter function of the vanilla GCN can be further approximated [1] as $g(\tilde{\lambda}_i) = 1 - \frac{\bar{p}}{\bar{p}+1} \tilde{\lambda}_i$, where $\bar{p}$ is the average node degree.

**Label Propagation (LP).** As the most classic graph-based semi-supervised learning method, label propagation [49] propagates label information from labeled data to unlabeled data along the graph. At the $k$-th iteration, it updates the predictive labels by $Y^{k+1} = \gamma D^{-\frac{1}{2}} A D^{\frac{1}{2}} Y^k + (1 - \gamma)Y$, where $\gamma$ is a hyper-parameter and $Y$ is a one-hot label matrix with setting $i$-th row to be zeros if $v_i$ is unlabeled. The fitted values of LP for all data are given in the closed form $Y^{lp} = (I + \frac{\gamma}{1-\gamma} L)^{-1} Y$, which also yields a low-pass filter [25] with $i$-th factor being reduced by the filter function $g(\lambda_i) = \frac{1}{1+a_1 \lambda_i}$ and $a_1 = \frac{\gamma}{1-\gamma}, a_1 \geq 0$.

**Ridge Regression.** Our proposed high-pass filter is deeply rooted in ridge regression. As a classic supervised model, ridge regression [9] shrinks the regression coefficients of the linear regression model by imposing a penalty on their size. Formally, given training data $X$ and corresponding labels $Y$, the least square solution for ridge regression, parameterized by $\beta$, is $\hat{\beta} = (X^T X + \lambda I)^{-1} X^T Y$, where $\lambda > 0$ is a hyper-parameter. The additional insight into the nature of $\hat{\beta}$ can be revealed by performing SVD on data $X = UDV^T$. It reveals that ridge regression shrinks the coordinates of $U$ by the factor $\frac{d_i^2}{(d_i^2+\lambda)}$, where $d_i \geq 0$ is the $i$-th singular value of $X$. Namely,

smaller $d_i$, corresponding to directions in the column space of $X$ having a smaller variance, suffer stronger shrinkage.

**Kernel Ridge Regression (KRR).** Following the Representer theorem [33], a model $f$ could be transformed into kernel expansion over training data, e.g., $f(x) = \sum_i^N \alpha_i K(x_i, x)$. Building upon this, the linear ridge regression mentioned above could be extended to the kernel ridge regression: $\hat{\alpha} = \arg\min_\alpha \|Y - K\alpha\|_2^2 + \lambda \alpha^T K\alpha$, from which we derive the fitted values as $K\hat{\alpha} = K(K + \lambda I)^{-1} Y = \Gamma(K, \lambda)Y$. Building upon the eigen-expansion of kernel $K = U_k \Lambda U_k^T$ with $\Lambda_{ii} \geq 0$ being the $i$-th eigenvalue of $K$, kernel ridge regression also puts fewer penalties on large eigenvalues in the eigen-expansion (or so-called *spectral decomposition*) of kernel $K$, with the $i$-th factor being $\frac{\lambda_i}{(\lambda_i + \lambda)}$. Note that $\Lambda_{ii} = \lambda_i$. Therefore, the fitted values of kernel ridge regression shrink by a high-pass spectral filter with the filter function $g(\lambda_i) = \frac{\lambda_i}{(\lambda_i + \lambda)}$.

## 4 CROSS-SPACE FILTER

In this section, we elaborate on CSF. It first leverages the correlations of node attributes to extract the interpretable high-pass filter arising from a Mercer's kernel (cf. Section 4.1). Then, it casts the conventional low-pass topology-based filter into another kernel, unifying the two filters in RKHS (cf. Section 4.2). Finally, the cross-space adaptive filter is obtained by applying a simple multiple-kernel learning technique to fuse the information in both the topology and attribute spaces (cf. Section 4.3).

### 4.1 High-pass Filter From Node Attribute Space

In deep GCNs, node representations in the whole graph get similar to each other and finally converge towards the same constant value. To this end, we aim to design a high-pass filter based on the node attributes to provide prior knowledge on which nodes should have dissimilar representations. To extract a filter from the attribute space, one straightforward idea is to create an attribute-based graph[1] and implement the GCN convolution operation. However, the challenge lies in how to extract a high-pass filter, especially in an interpretable way. In this paper, we resort to the KRR (cf. Section 3), which provides a rough idea for constructing a high-pass filter using node attributes in the graph. However, the general setting of the learning paradigm of GCNs is semi-supervised, where only partially labeled data are provided. In this section, we solve this challenge by solving an optimization problem of semi-supervised KRR and deriving an interpretable high-pass filter.

**Solving Semi-supervised KRR.** Let $\mathbb{Y} = \begin{bmatrix} Y_L \\ Z \end{bmatrix}$ denote the one-hot label matrix of all nodes, where $Z$ and $Y_L$ denote the labels of unlabeled and labeled nodes, respectively. The semi-supervised KRR is formulated as the following optimization problem.

$$\min_{\alpha, Z} \|\mathbb{Y} - K\alpha\|_2^2 + a_3 \alpha^T K\alpha + a_2 \|Z\|_2^2, \quad (1)$$

where $K$ is a kernel matrix over node attributes, and two regularization parameters $a_3$ and $a_2$ are introduced to control the complexity of model/hypothesis $f$ and the uniform prior on pseudo-labels $Z$, respectively. Although the problem is non-convex, it can still be approximated by a closed-form solution. In particular, denoting

---

[1]An attribute-based graph is obtained by calculating similarities of node attributes.

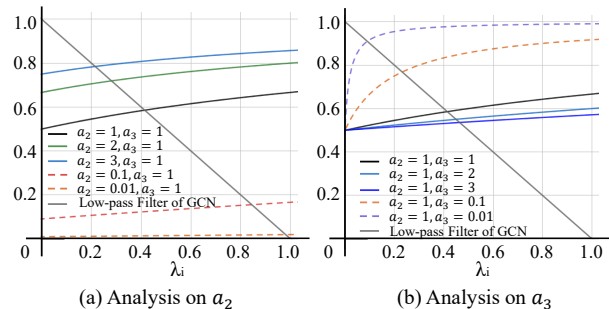

Figure 1: Overview of CSF. We leverage both the graph topology and node attribute spaces to produce a cross-space adaptive filter for alleviating the over-smoothing problem and improving the effectiveness of deep GCNs.

$\alpha = (a_3 I + K)^{-1} \mathbb{Y}$ as a variable parameterized by $a_3$, we plug it back into Problem (1) and yield the following penalized quadratic problem that only involves optimizing $Z$.

$$\min_Z \mathbb{Y}^T (I - \Gamma(K, a_3)) \mathbb{Y} + a_2 \|Z\|_2^2. \quad (2)$$

Notably, the transductive solution of the above optimization problem could be easily obtained in a closed form: $Z = -(\hat{K}_{uu} + a_2 I)^{-1} \hat{K}_{ul} Y_L$. We highlight that such a solution is the minimizer of the kernel regularized functional, taking the form of the regularized harmonic solution of label propagation [23].

**Deriving High-pass Filter.** Denote $Y_0$ as a one-hot label matrix, with the $i$-th row being zero if the $i$-th node in the graph is unlabeled. The problem 2 is translated as follows.

$$\min_{\mathbb{Y}} \mathbb{Y}^T (I - \Gamma(K, a_3)) \mathbb{Y} + a_2 \|Y_0 - \mathbb{Y}\|_2^2. \quad (3)$$

Consequently, the fitted value of semi-supervised KRR is derived as $\mathbb{Y} = K_{attr} Y_0$, where $K_{attr} = (I + \frac{1}{a_2}\hat{K})^{-1}$ and $\hat{K} = I - \Gamma(K, a_3)$.

PROPOSITION 1. $\hat{K} = I - \Gamma(K, a_3)$ is a valid kernel if and only if $a_3 > 0$. Also, $K_{attr}$ is a valid kernel if and only if $a_3 > 0$ and $a_2 > 0$.

PROOF. Note that $\hat{K} = 1 - \Gamma(K, a_3) = 1 - K(K + a_3 I)^{-1}$. We apply the SVD to the kernel $K = U_k \Lambda U_k^T$ and bring it back to $\hat{K}$, and we have the following equation holds.

$$\hat{K} = 1 - U_k(\Lambda(\Lambda + a_3 I)^{-1})U_k^T = \sum_i (1 - \frac{\lambda_i}{\lambda_i + a_3}) v_i^T v_i, \quad (4)$$

where $v_i$ and $\lambda_i$ correspond to the eigensystems of $K$ after SVD. Note that $\lambda_i > 0$ holds as $K$ is a valid kernel. Then, $\hat{K}$ is a valid kernel if $a_3 > 0$. This is because $1 - \frac{\lambda_i}{\lambda_i + a_3} = \frac{a_3}{\lambda_i + a_3} \geq 0$ holds for every eigenvalue of kernel $K$, as long as $a_3 > 0$ holds. Similarly, together with the SVD operation, we also conclude that $K_{attr} = (I + \frac{1}{a_2}\hat{K})^{-1}$ is a valid kernel if $a_2 > 0$ and $a_3 > 0$. □

PROPOSITION 2. The fitted values of semi-supervised kernel ridge regression are shrunk by a high-pass spectral filter, with the $i$-th factor being $g(\lambda_i) = \frac{a_2(\lambda_i + a_3)}{a_3 + a_2(\lambda_i + a_3)}$, where $a_2 > 0$, $a_3 > 0$, and $\lambda_i$ is the eigenvalue of kernel matrix $K$.

(a) Analysis on $a_2$   (b) Analysis on $a_3$

Figure 2: Illustration of filter function $g(\lambda_i)$ of $K_{attr}$. This demonstrates that $K_{attr}$ corresponds to a high-pass filter.

PROOF. The fitted values $\mathbb{Y} = K_{attr} Y_0 = (I + \frac{1}{a_2}\hat{K})^{-1} Y_0$ are shrunk by a low-pass filter with factor $\frac{a_2}{a_2 + \hat{\lambda}_i}$, where $\hat{\lambda}_i$ is the eigenvalue of $\hat{K}$. More importantly, $\mathbb{Y}$ is also shrunk by a high-pass spectral filter with factor $g(\lambda_i) = \frac{a_2(\lambda_i + a_3)}{a_3 + a_2(\lambda_i + a_3)}$, where $\lambda_i$ is the eigenvalue of kernel matrix $K$ that is derived from node attributes. □

Propositions 1 and 2 show that we could extract high-pass frequency information about node attributes by solving the semi-supervised KRR. Unlike the Laplacian matrix $\tilde{L}$ used in GCN's convolutional operation to extract low-pass topological information, $K_{attr}$ extracts high-pass attribute-based information.

REMARK 1. The kernel matrix $K_{attr}$ adjusts the shrinkage effect on the low-frequency signals in the attribute-based graph via two hyper-parameters, $a_2$ and $a_3$.

While $a_2 > 0$ and $a_3 > 0$ are the hyper-parameters for the regularization terms in Problem 1, they, as shown in Figure 2, control the shrinkage effect on the low-frequency signals in the attribute-based graph. Specifically, when $a_2$ or $a_3$ take very large values, our high-pass filter would become an all-pass filter[2], placing equal importance to both low-frequency and high-frequency signals in the

_____________
[2]When $a_2 \to \infty$ or $a_3 \to 0$, $g(\lambda_i) \to 1$. When $a_3 \to \infty$, $g(\lambda_i) \to \frac{a_2}{a_2 + 1}$

graph. On the one hand, $a_2$ controls the shrinkage strength, which is used to compress the scale of node attributes/representations. The smaller $a_2$, the stronger the compression ability. The information of node attributes would be lost when $a_2$ is very small (e.g., $a_2 = 0.01$). This property inspired us to take different values of $a_2$ when dealing with the assortative and disassortative graphs in the experiments[3], where the values of $a_2$ on disassortative graphs should be large. This highlights the importance of node attributes in multi-layer convolutional operations. On the other hand, no matter what value $a_3$ takes, it always prefers high-pass signals. Unlike $a_2$, $a_3$ controls the frequency range of the low-pass signals that need to be shrunk. The smaller $a_3$, the narrower the frequency range. Note that when $a_3 \rightarrow 0$ or $a_3 \rightarrow \infty$, our high-pass filter would become an all-pass filter. Besides, $a_3$ also controls the complexity of the semi-supervised KRR model. Thus, $a_3$ should not be too large or too small. Therefore, these two parameters control the shrinkage effect of our filter $K_{attr}$ on low-frequency signals in the attribute-based graph. It is worth noting that our filter remains a high-pass filter, regardless of the values $a_2$ and $a_3$ take. In some extreme cases, it acts as an all-pass filter, but it never becomes a low-pass filter. Refer to Appendix C for experimental analysis.

To sum up, the proposed spectral filter allows us to capture the correlations among node attributes and extract high-frequency information from the attribute space. Moreover, the derived filter is interpretable, as it's the minimizer of the semi-supervised kernel ridge regression problem.

## 4.2 Low-pass Filter From Graph Topology Space

The conventional low-pass filter of GCN is defined based on the Fourier graph. However, our proposed high-pass filter is defined based on Mercer's kernel. Thus, it would be challenging to combine these two filters and utilize their benefits simultaneously. To this end, we drew inspiration from the previous literature [36], which links the normalized Laplacian and Mercer's kernels on the graph. We show how to cast the low-pass filter as a Mercer's kernel in the context of GCNs, unifying the two filters in RKHS.

Specifically, let $r(e)$ be a Laplacian regularization function that monotonically increases in $e$ and that $r(e) \geq 0$ holds for all $e \in [0, 2]$, and $\{(e_i, \phi_i)\}$ be the eigensystem of a normalized Laplacian matrix, where $e_i$ and $\phi_i$ are the $i$-th eigenvalue and eigenvector. [36] shows that a kernel can be defined as $K = \sum_{i=1}^{m} r^{-1}(e_i)\phi_i\phi_i^T$, where $r^{-1}$ is the spectral filter function[4]. In the context of GCNs, we define a specified Laplacian regularization function as $g_1^{-1}$ for GCN, where $g_1(e) = 1 - \frac{\bar{p}}{\bar{p}+1}e$ is the filter function of GCN discussed in Section 3. In this case, the filters of GCN are cast into the Mercer's kernel space, where the low-pass filter of GCN is the following one-step random walk kernel $K^{gcn} = I - \frac{\bar{p}}{\bar{p}+1}\tilde{L}$ with a spectral filter on the eigenvalues of the graph Laplacian, $r(\tilde{\lambda}_i) = \frac{\bar{p}+1}{\bar{p}(1-\tilde{\lambda}_i)+1}$.

## 4.3 Integrating Topology-based and Attribute-based Filters

To solve the problem of overlooking node attributes in existing adaptive filter methods, we aim to integrate topology and attribute

---

**Algorithm 1** Pseudo-code of CSF

---

**Input:** Graph $G$ and node attribute matrix $X$. Maximum epoch $EP = 150$.

**Default Parameters:** Top-k = 20 for KNN graph. $a_3 = 1$, $a_2 = 1$. Refer to [4, 5], $\gamma$ is selected by cross-validation.

1: %%% *Obtain high-pass filter from attribute space.* %%%
2: Construct a Gaussian kernel $K$ via KNN graph on $X$.
3: $\Gamma(K, a_3) = K(K + a_3 I)^{-1}$
4: $K_{attr} = (I + \frac{1}{a_2}(I - \Gamma(K, a_3)))^{-1}$.
5: %%% *Obtain low-pass filter from topology space via GCN.* %%%
6: $K_{top} = I - \tilde{L} = \tilde{D}^{-\frac{1}{2}}\tilde{A}\tilde{D}^{-\frac{1}{2}}$
7: %%% *Obtain cross-space adaptive filter by MKL [4, 5].* %%%
8: $\mathbb{K} = (K_{attr} + K_{top})/2 + \gamma(K_{attr} - K_{top})(K_{attr} - K_{top})$.
9: %%% *Perform propagation* %%%
10: Initialize $epo = 0$, $k = 0$.
11: Set $H^k = X$, and diagonal matrix $\hat{D}$, with $\hat{D}_{ii} = \sum_j \mathbb{K}_{ij}$.
12: **while** $EP > epo$ **do**
13: $\quad H^{k+1} = \sigma(\hat{D}^{-\frac{1}{2}}\mathbb{K}\hat{D}^{-\frac{1}{2}}H^kW^k) \oplus X$
14: $\quad epo+=1$
15: **end while**

---

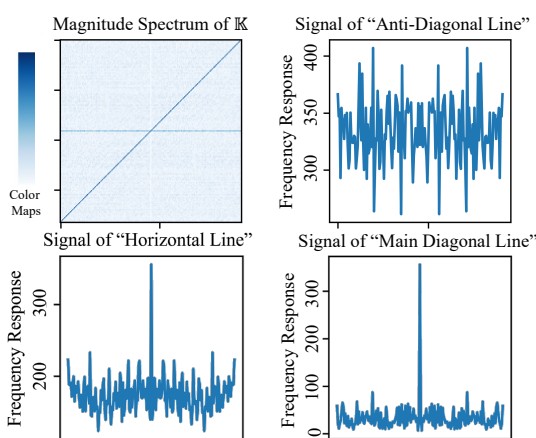

**Figure 3: Illustration of magnitude spectrum and frequency response. This demonstrates that $\mathbb{K}$ corresponds to an adaptive filter that combines both low-pass and high-pass filters.**

information from the perspective of Mercer's kernel. As shown in Figure 1, our adaptive filter CSF is obtained by applying a simple yet effective multiple-kernel learning technique to fuse the information. We elaborate on our CSF below.

**Obtain Attribute-based Filter**. Given training data with limited labels, we first construct a kernel based on the node attribute $X$ via the k-nearest-neighbor (KNN) graph[5], whose edges are weighted by the Gaussian distance. This forms a valid Gaussian kernel $K$. To enhance the optimization stability, we also exploit the re-normalization and self-loop tricks on the kernel $K$, which are used in vanilla GCN. Next, we calculate $K_{attr}$ to build the high-pass spectral filter and capture the information of the node attributes.

**Obtain Topology-based Filter**. The one-step random walk kernel from vanilla GCN is utilized, which we found to be highly

---

[3]Refer to Foster et al. [7] and Networkx package to check if a graph is disassortative.
[4]Note that the pseudo-inverse and $0^{-1} = 0$ are applied wherever necessary.

[5]Refer to Appendix B for the robustness analysis on the KNN graph.

effective. In our experiments, we resort to the raw propagation form of GCN and set $K_{top} = I - \tilde{L} = \tilde{D}^{-\frac{1}{2}} \tilde{A} \tilde{D}^{-\frac{1}{2}}$. We call attention to the importance of node attributes to the over-smoothing problem of GCN, and we will defer the research question of proposing a new topology-based filter to our future work.

**Obtain Cross-Space Filter.** To fuse the information across both graph topology and node attribute spaces, a multiple kernel learning (MKL) algorithm is adopted. Inspired by previous work, a *squared matrix-based MKL* [4, 5, 14] is used to integrate topology and attribute information from the perspective of Mercer's kernel.

$$\mathbb{K} = \frac{K_{attr} + K_{top}}{2} + \gamma (K_{attr} - K_{top})(K_{attr} - K_{top}). \quad (5)$$

Referring to [4, 5], the second term in $\mathbb{K}$ represents the difference of information between $K_{attr}$ and $K_{top}$, and $\gamma$ is a positive constant selected by cross-validation used to control the relative importance of this difference. Other advanced MKL methods may also work, but that is not the research focus of this paper. In this paper, we highlight that this squared matrix-based algorithm is effective and simple without trainable parameters. It not only avoids the computational overhead of parameter learning and improves time efficiency, but also shows the capability of leveraging the advantages of two spaces (cf. Section 5). Importantly, the following corollary shows that Eq.5 produces a valid kernel matrix $\mathbb{K}$, which is the foundation of being a valid filter on the graph according to Smola and Kondor [36].

COROLLARY 1. $\mathbb{K}$ *is a positive definite kernel matrix.*

PROOF. Apparently, the sum of two kernels yields a kernel. Therefore, the first term $\frac{K_{attr} + K_{top}}{2}$ is a kernel. Let $K_{at} = K_{attr} - K_{top}$. $K_{at}$ is symmetric since both $K_{attr}$ and $K_{top}$ are symmetric. Then, there exists an orthogonal matrix $Q_{at}$ such that $K_{at} = Q_{at}^T \Lambda_{at} Q_{at}$, where $\Lambda_{at}$ is a diagonal matrix whose elements are the eigenvalues of $K_{at}$. Now $K_{at}^2 = Q_{at}^T \Lambda_{at}^2 Q_{at}$, that is, $K_{at}^2$ is positive semidefinite. Taking the first and second terms together, Eq.5 provides a matrix $\mathbb{K}$ arising from a Mercer's kernel. Then $\mathbb{K}$ could play the role of a valid filter on the graph, according to Smola and Kondor [36].  □

REMARK 2. $\mathbb{K}$ *corresponds to an adaptive filter that combines both low-pass and high-pass filters.*

To illustrate the adaptive filter, we plot the magnitude spectrum and frequency response of $\mathbb{K}$ on the Texas dataset in Figure 3. We selected three representative signals based on the magnitude spectrum results. The frequency response of these signals indicates that $\mathbb{K}$ successfully integrates the low-pass (cf. the main diagonal and horizontal lines) and high-pass filters[6] (cf. the anti-diagonal line).

**Propagation rule of CSF.** Let $\hat{D}_{ii} = \sum_j \mathbb{K}_{ij}$ and $H^0 = X$. The layer-wise propagation rule of our CSF method is concluded as $H^{k+1} = \sigma(\hat{D}^{-\frac{1}{2}} \mathbb{K} \hat{D}^{-\frac{1}{2}} H^k W^k) \oplus X$. Taking inspiration from the propagation rule of the label propagation algorithm, which attaches the response variable $Y$ to the propagation to ensure consistency between predicted labels and $Y$, we also attach raw attributes $X$ to the propagation process to enhance consistency with the raw feature $X$. Here, $\oplus$ means the concatenation operator.

---

[6]This high-pass filter is less similar to an all-pass filter as the frequency response in the middle part is still lower than the other parts on the line.

We conclude the pseudo-code of CSF in Algorithm 1. Unlike GAT and GCN-BC, which suffer from memory overflow due to their high space complexity, our method requires a time complexity of $O(N^3)$ to obtain the cross-space filter due to kernel inversion. Fortunately, 1) we only calculate the kernel once (see Algorithm 1), 2) classic methods, such as the Nystrom method [27, 41] can help ease the calculation. In Appendix D, we reduce the complexity of the kernel inversion to $O(mN^2)$, where $m \ll N$, which is comparable to the filter construction of vanilla GCN (i.e., $O(N^2)$).

## 5 EMPIRICAL EVALUATION

To assess CSF's ability to mitigate the over-smoothing issue and promote the effectiveness power of deep GCN, we conduct comparative experiments with various baselines under different numbers of convolution layers (see Section 5.2). Moreover, we reveal more characteristics of CSF by conducting ablation studies on graph topology and node attributes (see Section 5.3).

### 5.1 Experiment Setup

**Datasets.** Following previous studies on adaptive filters [2, 19], nine commonly used node classification datasets are utilized. This includes three assortative graphs (i.e., Cora, Citeseer, and Pubmed) and six disassortative graphs (i.e., Actor, Cornell, Texas, Wisconsin, Chameleon, and Squirrel). In an assortative graph, nodes tend to connect with other nodes bearing more similarities, while ones in the disassortative graph are on the contrary. All datasets are available online. Refer to Bo et al. [2] for more details on the datasets.

**Comparative Methods.** Focusing on the over-smoothing problem, we rely on adaptive filter methods as our primary baselines. In particular, we consider FAGCN [2] and PGNN [10] which integrate low-pass and high-pass filters, AKGNN [19] which integrates all-pass and low-pass filters, and AdaGNN [6] which has a trainable filter function. Additionally, we take representative GNNs into comparison, including GCN [22], SGC [42], GAT [39], GraphSage [15]. To evaluate the effectiveness of node attributes, we compare with attribute enhanced GNNs, including APPNP [13], JKNet [44] and GCN-BC [46]. Finally, MLP is used as our supervised baseline, which is trained using the node attributes.

**Implementation Details.** The codes for most baseline methods are publicly available online. However, since there is no code available for GCN-BC, we implemented this method based on the propagation rules mentioned in their paper. Regarding the MLP, each layer is implemented by a fully connected network with ReLu activation and dropout. In the case of our proposed CSF, we utilized node attributes to construct a KNN graph with the top-20 neighbors[7]. To calculate edge similarity among nodes, we used a Gaussian kernel with the kernel bandwidth set to the squared average distance among nodes, without further tuning. We set $a_2 = 0.1$ for the assortative graphs, $a_2 = 100$ for the disassortative graphs, and $a_3 = 1$ for all graphs (cf. Appendix C for the analysis on $a_2$ and $a_3$). To simulate the deep GNNs, we tune the number of convolution layers of each method (i.e., {2, 5, 10, 20}). Moreover, unlike FAGCN[2], we do not tune the hidden size on each dataset. Instead, we followed previous work [20, 26, 40] and set the hidden size to 16 for all methods. We

---

[7]Refer to Appendix B for the robustness analysis on the KNN graph.

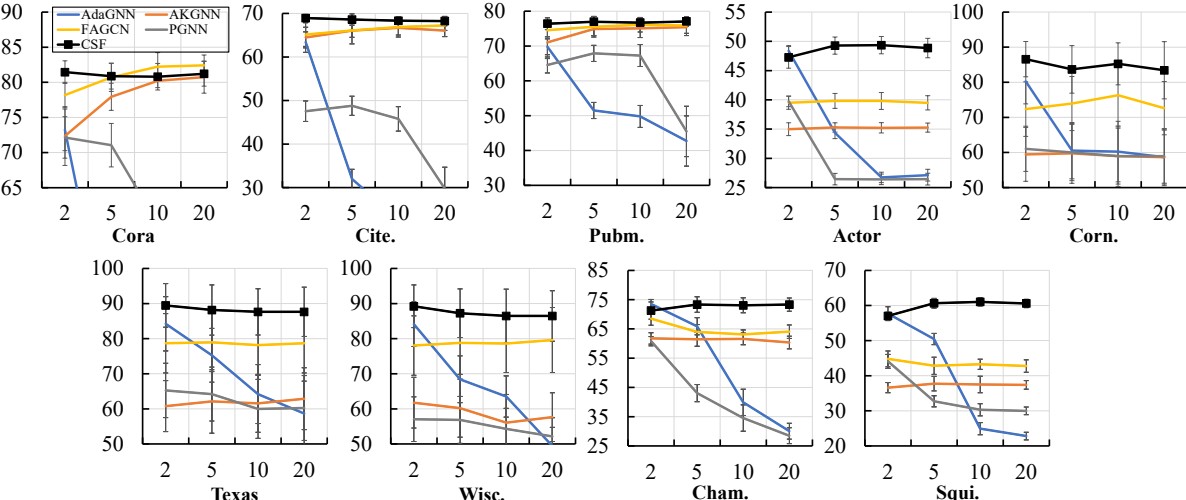

Figure 4: Over-smoothing problem evaluation of adaptive filter-based methods. We tune the number of layers of each method (i.e., the X-axis). The Y-axis represents the classification accuracy (%). This indicates that CSF outperforms others in terms of its robustness to over-smoothing problem and its effectiveness on downstream tasks.

Table 1: Performance of all methods for node classification averaged over different numbers of convolution layers (i.e., $\{2, 5, 10, 20\}$). The "*" symbol indicates that we ignored the memory overflow situation of the corresponding model. The best results are marked in bold and the second best are underlined. For more detailed results, please refer to Table 6 in the Appendix. The results indicate CSF improves the expressiveness of deep GCN, especially when dealing with disassortative graphs.

| Model | Assortative Graphs (%) | | | Disassortative Graphs (%) | | | | | |
|---|---|---|---|---|---|---|---|---|---|
| | Cora | Cite. | Pubm. | Actor | Corn. | Texas | Wisc. | Cham. | Squi. |
| MLP | 37.46±0.62 | 30.54±1.20 | 50.71±0.90 | 34.54±0.36 | 67.96±0.90 | 68.16±0.15 | 62.75±0.37 | 36.50±2.01 | 29.41±2.73 |
| GCN | 48.85±2.25 | 40.15±1.66 | 55.77±1.43 | 27.63±0.16 | 59.60±0.32 | 61.45±0.01 | 51.72±1.63 | 34.51±4.81 | 23.90±0.63 |
| SGC | 65.56±1.45 | 52.76±1.29 | 71.11±1.44 | 28.51±0.25 | 61.19±0.69 | 63.03±0.74 | 55.15±0.73 | 45.85±0.51 | 28.10±0.72 |
| GAT | 64.01±0.42 | 58.88±2.66 | 74.84±2.01* | 30.58±0.16 | 63.42±1.04 | 66.65±1.07 | 60.15±0.48 | 60.89±0.23 | 37.68±0.20 |
| GraphSAGE | 45.26±0.48 | 41.97±0.39 | 60.34±0.61 | 30.19±0.09 | 59.41±0.19 | 61.78±1.31 | 56.91±1.08 | 55.14±0.24 | 32.48±0.36 |
| APPNP | 78.37±0.24 | 68.16±0.35 | 75.40±0.11 | 27.04±0.19 | 59.34±0.21 | 60.39±0.26 | 53.19±0.88 | 39.11±0.23 | 21.01±0.32 |
| JKNET | 76.44±0.30 | 65.47±0.30 | 74.80±0.11 | 26.75±0.55 | 59.34±0.33 | 60.99±0.41 | 56.76±0.96 | 43.94±0.71 | 30.31±0.33 |
| GCN-BC | 23.39±2.38 | 19.21±2.05 | 73.36±2.88* | 15.20±0.18 | 66.05±0.58 | 67.50±0.57 | 72.21±0.85 | 33.14±0.55 | 27.94±0.25 |
| FAGCN | 80.88±0.25 | 66.35±0.67 | 75.51±0.44 | 39.66±0.14 | 73.82±0.68 | 78.62±0.11 | 78.78±0.45 | 64.91±0.44 | 43.41±0.45 |
| PGNN | 60.22±1.83 | 42.98±1.26 | 61.30±2.35 | 29.76±0.06 | 59.74±0.93 | 62.43±0.73 | 55.10±0.59 | 41.72±1.20 | 34.27±0.38 |
| AdaGNN | 44.53±0.58 | 34.80±0.89 | 53.45±1.22 | 34.15±0.24 | 64.93±0.94 | 70.59±2.08 | 66.37±1.55 | 52.34±0.58 | 38.97±0.74 |
| AKGNN | 77.81±1.35 | 65.84±0.69 | 74.10±1.22 | 35.19±0.14 | 59.21±0.35 | 61.84±0.79 | 58.92±0.64 | 61.28±0.20 | 37.31±0.52 |
| CSF | **81.09±0.12** | **68.54±0.31** | **76.79±0.30** | **48.68±0.18** | **84.74±1.32** | **88.22±0.43** | **87.35±0.66** | **72.74±0.29** | **59.82±0.14** |
| Improvement | 0.21 ↑ | 0.38 ↑ | 1.28 ↑ | 9.02 ↑ | 10.92 ↑ | 9.6 ↑ | 8.57 ↑ | 7.83 ↑ | 16.41 ↑ |

also note that the learning rate in previous studies is either too large (e.g., [24] directly uses $\{0.5\}$) or too small (e.g., [2] tunes the learning rate in $\{0.01, 0.005\}$). Therefore, we tune the learning rate to fully evaluate the performance of all methods when the learning rate is large or small, namely $\{0.03, 0.02, 0.01, 0.005, 0.1, 0.2, 0.3, 0.4, 0.5\}$. Finally, all experiments were conducted on a machine with an Intel(R) Core(TM) i5-12400F, 16GB memory, and GeForce RTX 3060. We implemented all methods in Pytorch and optimized them using the Adam optimizer with a dropout rate of 0.5. Finally, we ran 150 epochs and selected the model with the highest validation accuracy for testing. We report the mean and variance of the results over ten runs, where all experiments were conducted under the same fixed

random seed to ensure that different methods were performed with the same labeled data in each run.

## 5.2 Main results

Previous works show vanilla GCN suffers from the over-smoothing problem once the network goes deep. In this section, we design experiments to evaluate the performances of all methods on various datasets, where the number of layers ranges from $\{2, 5, 10, 20\}$. Due to space limitations, we report detailed results in the Appendix (cf. Table 6). Our experimental results, as shown in Fig. 4 and Table 1, indicate that **our alleviates the over-smoothing problem while**

**at the same time promoting the effectiveness of deep GCNs especially when dealing with disassortative graphs.**

Specifically, Fig. 4 illustrates the performance of CSF and existing adaptive filter methods in terms of alleviating the over-smoothing problem. The results indicate that CSF remains effective even with increasing model layers and shows a trend of increasing effectiveness on some datasets (such as Cham., Squi., Pumb.). In contrast, we observe that AdaGNN and PGNN may not fully address the over-smoothing issue as their effectiveness tends to decrease with the increase of model layers. Compared to the remaining two baselines (i.e., FAGNN and AKGNN), although CSF only achieves comparative performance on Cora data, it outperforms FAGNN and AKGNN in most cases, especially when dealing with the disassortative graphs. These results highlight the importance of alleviating the over-smoothing problem while at the same time promoting the effectiveness of deep GCNs. For more quantified results, we summarize the experimental results of all models in Table 1, averaged over the number of layers. The results indicate that CSF significantly outperforms other comparative methods on all datasets, especially on disassortative graphs. This aligns with previous research that suggests GCN and its variants perform poorly on the disassortative graphs [18] because their nodes tend to connect to others with dissimilar properties, making topology information unreliable for downstream tasks. Therefore, combining information from both topology and attribute is necessary. Although current works attempt to integrate both factors, such as JKNet's jumping knowledge, or APPNP's and GCN-BC's hidden representation to improve representation capabilities, they still underperform CSF. This implies that extracting information from node attributes in the form of a high-pass filter is a better solution. It not only enhances model performance but also alleviates the over-smoothing problem.

## 5.3 Ablation studies

We further conduct an in-depth analysis to reveal the characteristics of node attributes, as illustrated in Fig. 5.

**For assortative graphs, topology information is more important, whereas, for disassortative graphs, node attribute information is more valuable. CSF effectively balances the two**. To evaluate the impact of different spaces, we eliminate the attribute filter (i.e., *CSF -w/o attribute*) and topology filter (i.e., *CSF -w/o topology*) from $\mathbb{K}$ individually. The results in Fig. 5(a) show that topology information dominates attribute information on assortative graphs, while the opposite is true for disassortative graphs. This leads to two observations: firstly, our proposed high-pass filter successfully extracts attribute information, leading to better performance on disassortative graphs than topology-based filters. Secondly, the MKL method used in CSF integrates the advantages of both spaces and achieves better overall performance.

**High-frequency information in attribute space is more valuable than low-frequency information**. We first introduce a baseline *CSF -w low-pass attribute* that involves replacing the high-pass filter $K_{attr}$ with the vanilla GCN filter performed on the attribute-based graph. In this case, two low-pass filters from different spaces are used to build $\mathbb{K}$. For comparison, we then introduce a baseline *CSF -w only low-pass attribute* that uses the same low-pass attribute-based filter alone as $\mathbb{K}$. As demonstrated in Figure 5(b), it leads to two observations. 1) Comparing *CSF -w low-pass*

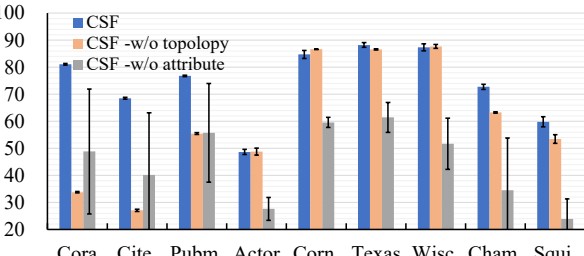

**(a) Ablation on topology and attribute spaces**

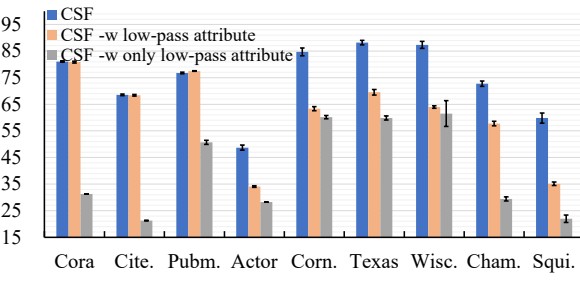

**(b) Ablation on filters from attribute space**

**Figure 5: Ablation studies on node attribute space. We report the averaged performance across various number of layers (i.e., Y-axis). Fig.(a) shows CSF effectively balances the advantages of two spaces, while Fig.(b) highlights the importance of high-frequency information in attribute space.**

*attribute* with *CSF -w only low-pass attribute* and *CSF -w/o attribute*, fusing filters from two spaces still has an effective gain even if they are all low-pass filters. 2) Comparing CSF with *CSF -w low-pass attribute*, it highlights the high-frequency information in attribute space, securing the superiority of our CSF.

We also analyze the parameter sensitivity and robustness of $K_{attr}$ in the Appendix C and Appendix B, respectively.

## 6 CONCLUSION

Alleviating the over-smoothing problem while at the same time promoting the effectiveness power is known to be important for applying deep GCNs to downstream tasks. Existing methods fail on this challenge due to heavily relying on graph topology and overlooking the correlation information in node attributes. To torch this challenge, we take the first step to propose a high-pass attribute-based filter, which is interpreted as a minimizer of semi-supervised kernel ridge regression. More importantly, for the first time, we propose a cross-space adaptive filter arising from a Mercer's kernel. Such a filter integrates information across both the topology and attribute spaces, resulting in superior robustness to the over-smoothing problem and promoting the effectiveness of deep GCN on downstream tasks, as demonstrated in our experiments.

We would like to emphasize that our proposed method provides a new perspective for the GCN community. It provides insight into revisiting the role of node attributes and kernels in alleviating the over-smoothing problem. Additionally, its potential value lies in offering a more convenient and interpretable way to design customized spectral filters and integrate them together, regardless of their space sources.

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

**Table 2: Ablation on the impact of different spaces.**

| Method/Graphs | # Layers | | | |
|---|---|---|---|---|
| | 2 | 5 | 10 | 20 |
| **Cora** | | | | |
| CSF | **81.45±1.61** | **80.88±1.85** | **80.8±1.89** | **81.21±1.75** |
| CSF -w/o topology | 33.60±1.39 | 34.09±1.15 | 33.88±1.01 | 33.5±1.27 |
| CSF -w/o attribute | 80.03±2.02 | 52.78±5.52 | 31.3±0.76 | 31.3±0.76 |
| **Cite.** | | | | |
| CSF | **68.94±0.84** | **68.61±1.31** | **68.33±0.85** | **68.26±1.08** |
| CSF -w/o topology | 27.22±1.67 | 26.52±2.24 | 27.51±1.74 | 27.16±2.06 |
| CSF -w/o attribute | 67.79±1.22 | 50.43±4.50 | 21.53±2.02 | 20.83±0.78 |
| **Pubm.** | | | | |
| CSF | **76.41±1.70** | **76.98±1.54** | **76.72±1.41** | **77.06±1.35** |
| CSF -w/o topology | 55.39±2.91 | 55.14±4.24 | 55.9±2.36 | 55.47±3.3 |
| CSF -w/o attribute | 75.01±1.88 | 67.68±3.74 | 40.49±1.92 | 39.88±0.25 |
| **Actor** | | | | |
| CSF | 47.26±1.85 | 49.27±1.46 | 49.34±1.48 | 48.86±1.66 |
| CSF -w/o topology | 46.89±0.8 | **49.54±0.57** | **49.38±0.66** | **49.49±0.85** |
| CSF -w/o attribute | 33.99±1.03 | 25.7±1.24 | 25.41±1.37 | 25.41±1.37 |
| **Corn.** | | | | |
| CSF | 86.58±5.03 | 83.68±6.77 | 85.26±5.98 | 83.42±8.14 |
| CSF -w/o topology | **86.84±7.13** | **86.58±6.96** | **86.58±6.26** | **86.58±6.73** |
| CSF -w/o attribute | 62.37±8.78 | 58.68±8.14 | 58.68±8.14 | 58.68±8.14 |
| **Texas** | | | | |
| CSF | **89.47±6.2** | **88.16±7.15** | **87.63±6.57** | **87.63±7.02** |
| CSF -w/o topology | 86.58±6.26 | 86.58±6.50 | 86.84±6.56 | 86.32±6.88 |
| CSF -w/o attribute | 69.74±8.16 | 58.68±8.14 | 58.68±8.14 | 58.68±8.14 |
| **Wisc.** | | | | |
| CSF | **89.22±6.08** | **87.25±6.93** | 86.47±7.65 | 86.47±7.19 |
| CSF -w/o topology | 88.63±6.39 | **87.25±6.75** | **87.84±6.46** | **87.06±6.61** |
| CSF -w/o attribute | 65.88±8.63 | 48.04±4.74 | 46.47±6.06 | 46.47±6.06 |
| **Cham.** | | | | |
| CSF | **71.27±2.95** | **73.33±2.65** | **73.05±2.54** | **73.31±2.26** |
| CSF -w/o topology | 62.94±2.30 | 63.51±1.85 | 63.29±2.08 | 63.38±1.73 |
| CSF -w/o attribute | 62.46±2.32 | 32.61±10.88 | 21.45±0.89 | 21.51±0.84 |
| **Squi.** | | | | |
| CSF | **57.03±1.04** | **60.64±1.38** | **61.04±1.17** | **60.58±1.17** |
| CSF -w/o topology | 51.05±1.75 | 54.49±1.27 | 54.06±1.52 | 54.15±1.30 |
| CSF -w/o attribute | 34.97±2.32 | 20.12±1.10 | 20.17±1.04 | 20.32±1.03 |

**Table 3: Ablation on the impact of different filters from note attribute space.**

| Method/Graphs | # Layers | | | |
|---|---|---|---|---|
| | 2 | 5 | 10 | 20 |
| **Cora** | | | | |
| CSF | **81.45±1.61** | **80.88±1.85** | **80.8±1.89** | **81.21±1.75** |
| CSF -w low-pass attribute | 81.43±1.63 | 80.76±1.64 | 80.61±1.95 | 80.68±1.82 |
| CSF -w only low-pass attribute | 31.33±0.79 | 31.37±0.82 | 31.27±0.82 | 31.27±0.82 |
| **Cite.** | | | | |
| CSF | **68.94±0.84** | **68.61±1.31** | **68.33±0.85** | **68.26±1.08** |
| CSF -w low-pass attribute | 68.84±0.81 | 68.28±1.00 | 68.32±1.07 | 68.07±1.31 |
| CSF -w only low-pass attribute | 21.41±1.08 | 21.43±1.24 | 21.12±0.89 | 21.15±0.67 |
| **Pubm.** | | | | |
| CSF | **76.41±1.70** | **76.98±1.54** | 76.72±1.41 | 77.06±1.35 |
| CSF -w low-pass attribute | 77.4±1.86 | 77.64±1.21 | **77.41±1.07** | **77.52±2.07** |
| CSF -w only low-pass attribute | 50.18±0.83 | 49.88±0.77 | 51.1±0.89 | 51.56±0.57 |
| **Actor** | | | | |
| CSF | **47.26±1.85** | **49.27±1.46** | **49.34±1.48** | **48.86±1.66** |
| CSF -w low-pass attribute | 33.61±0.65 | 34.23±0.82 | 34.32±0.92 | 34.17±0.75 |
| CSF -w only low-pass attribute | 28.25±0.77 | 28.28±0.75 | 28.18±0.78 | 28.19±0.64 |
| **Corn.** | | | | |
| CSF | **86.58±5.03** | **83.68±6.77** | **85.26±5.98** | **83.42±8.14** |
| CSF -w low-pass attribute | 63.95±8.42 | 62.37±8.78 | 63.16±8.5 | 63.95±7.85 |
| CSF -w only low-pass attribute | 60.53±9.12 | 60.53±9.45 | 59.21±8.62 | 60.26±8.36 |
| **Texas** | | | | |
| CSF | **89.47±6.20** | **88.16±7.15** | **87.63±6.57** | **87.63±7.02** |
| CSF -w low-pass attribute | 70.00±7.67 | 68.68±8.18 | 70.79±8.55 | 68.68±8.81 |
| CSF -w only low-pass attribute | 60.53±8.95 | 58.95±7.96 | 59.47±9.3 | 60.53±8.86 |
| **Wisc.** | | | | |
| CSF | **89.22±6.08** | **87.25±6.93** | **86.47±7.65** | **86.47±7.19** |
| CSF -w low-pass attribute | 64.71±8.42 | 63.73±8.93 | 63.73±8.88 | 63.73±7.96 |
| CSF -w only low-pass attribute | 57.84±8.78 | 57.65±10.83 | 62.75±9.96 | 67.84±7.96 |
| **Cham.** | | | | |
| CSF | **71.27±2.95** | **73.33±2.65** | **73.05±2.54** | **73.31±2.26** |
| CSF -w low-pass attribute | 58.82±1.75 | 56.78±1.84 | 57.87±2.47 | 57.57±2.42 |
| CSF -w only low-pass attribute | 28.31±3.07 | 29.34±2.29 | 29.93±2.5 | 29.98±2.52 |
| **Squi.** | | | | |
| CSF | **57.03±1.04** | **60.64±1.38** | **61.04±1.17** | **60.58±1.17** |
| CSF -w low-pass attribute | 36.12±1.07 | 34.74±0.97 | 34.84±1.19 | 34.76±1.5 |
| CSF -w only low-pass attribute | 20.46±2.31 | 22.83±1.48 | 21.07±2.53 | 23.44±1.76 |

## A    APPENDIX FOR RELATED WORK

**Connection to Shrinkage Estimator**. In statistics, a shrinkage estimator is an estimator that incorporates the effects of shrinkage according to the extra information [8], such as the kernel matrix [29]. Classic examples include the Lasso estimator for Lasso regression, the ridge estimator for ridge regression, and the spectral kernel mean shrinkage estimator [29, 30] for kernel mean estimation. Here, our high-pass spectral filter is also a shrinkage estimator for semi-supervised kernel ridge regression, wherein the shrinkage strength is small in the coordinates with large eigenvalues.

**Graph Structure Learning.** It targets to simultaneously train an optimized graph topology and corresponding node embeddings for downstream tasks [52]. However, our CSF approach differs from graph structure learning in that we do not explicitly learn the structure of the graph. Instead, we focus on multiple kernel learning to integrate kernels from different spaces.

## B    ANALYSIS ON INITIALIZATION OF $K_{attr}$

Constructing a KNN-based graph is the most popular initialization method for the information propagation model [11], CSF also uses it as the initialization of the attribute-based filter $K_{attr}$ (i.e., the Gaussian kernel $K$). In this section, we evaluate the sensitivity of CSF concerning the sparsity of the KNN-based graph. We vary the number of neighbors of KNN in $\{5, 10, 20, 50\}$ and extract the high-pass filter as usual. The results, presented in Table 4, indicate that CSF is generally robust to the sparsity of KNN-based kernel initialization. However, we recommend using the top 20 neighbors to construct the KNN kernel in practice, as it leads to better overall performance than other options.

## C    HYPER-PARAMETER ANALYSIS

As discussed in Remark 1 in Section 4.1, two hyper-parameters, The kernel matrix $K_{attr}$ adjusts the shrinkage effect on the low-frequency signals in the attribute-based graph via $a_2$ and $a_3$. In particular, $a_2$ controls the shrinkage strength, which is used to compress the scale of node attributes/representations, while $a_3$

**Table 4: Robustness evaluation on KNN-based initialization. For simplicity, the experiments are conducted with ten layers.**

| #Neighbors | Assortative | | | Disassortative | | | | | |
|---|---|---|---|---|---|---|---|---|---|
| | Cora | Cite. | Pubm. | Actor | Corn. | Texas | Wisc. | Cham. | Squi. |
| Top 5 | 80.93±1.91 | 67.85±1.06 | 76.85±1.77 | 50.02±1.74 | 85.22±4.87 | 87.65±6.68 | 87.11±7.23 | 72.86±2.08 | 60.56±1.33 |
| Top 10 | 80.67±1.09 | 68.03±0.89 | 76.61±1.84 | 49.14±1.57 | 82.19±7.03 | 87.62±6.98 | 86.49±7.14 | 73.46±2.22 | 61.23±1.08 |
| Top 20 | 80.80±1.89 | 68.33±0.85 | 76.72±1.41 | 49.34±1.48 | 85.26±5.98 | 87.63±6.57 | 86.47±7.65 | 73.05±2.54 | 61.04±1.17 |
| Top 50 | 79.77±1.45 | 68.63±0.80 | 76.71±1.38 | 49.55±1.44 | 82.23±6.88 | 87.67±6.55 | 87.18±7.12 | 72.77±2.51 | 60.15±0.95 |
| Full connect | 74.76±2.49 | 69.23±0.65 | 76.84±1.33 | 49.13±1.40 | 80.16±7.77 | 87.60±6.53 | 87.47±8.08 | 73.95±2.14 | 60.04±1.04 |

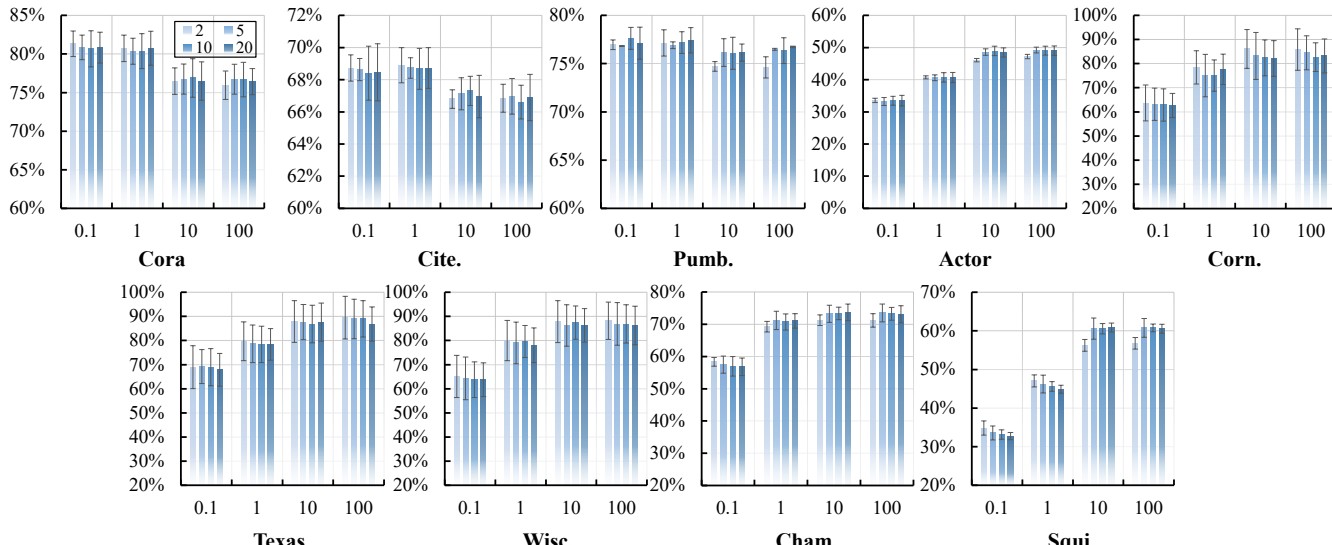

**Figure 6: Parameter analysis on $a_2$ on various number of layers ($\{2, 5, 10, 20\}$). Here $a_2$ is tuned in $\{0.1, 1, 10, 100\}$, and $a_3$ is fixed as 1. We report the average performance (and its standard deviation) of CSF across different numbers of layers, where the black line represents the standard variance.**

controls the frequency range of the low-pass signals that need to be shrunk. In this section, we experimentally analyze the sensitivity of CSF to these two hyperparameters. Fig. 6 shows the experimental results of parameter tuning on $a_2$, while Fig. 7 shows the results of tuning $a_3$. The detailed observations are as follows.

**Analysis on $a_2$.** According to Fig. 6, assortative graphs favor small $a_2$, while disassortative graphs favor large $a_2$. This finding is consistent with our analysis. $a_2$ controls the shrinkage strength, which is used to compress the scale of node attributes/representations. The smaller $a_2$, the stronger the compression ability. The information about node attributes would be lost when $a_2$ is very small. In this case, $a_2$ controls the importance of node attributes in downstream tasks. As for disassortative graphs, taking a larger value of $a_2$ helps to leverage the information of node attributes. Arbitrarily setting $a_2 = 1$ is a safe choice, but we suggest using the NetworkX package ( and partially labeled data) to check if a given graph is disassortative.

**Analysis on $a_3$.** According to Fig. 7, CSF is more robust to the value of $a_3$ compared to $a_2$, as $a_3$ only controls the frequency range of the low-pass signals that need to be shrunk. However, $a_3$ also controls the complexity of the semi-supervised KRR model. Thus,

$a_3$ should not be too large or too small. This may explain why the performance of CSF decreases on the Actor dataset. In this paper, we suggest setting $a_3 = 1$.

## D IMPROVING THE COMPUTATION EFFICIENCY OF KERNEL INVERSION

Our proposed approach effectively addresses the over-smoothing problem of GCN and performs well compared to various baselines on different datasets. Here, we discuss the potential limitations of computational complexity. Unlike GAT and GCN-BC, which suffer from memory overflow due to their high space complexity, our method requires high time complexity to obtain the cross-space filter. Normally, it requires $O(N^3)$ due to kernel multiplication and inversion. Fortunately, 1) we only calculate the kernel once (see Algorithm 1), 2) and classic methods, such as the low-rank approximation [16], random Fourier feature [3, 31] and Nystrom method [27, 41] can help ease the calculation on large graphs.

To demonstrate this, we tentatively provide a proof-of-concept experiment on Nystrom-based CSF. This reduces the complexity to $O(mN^2)$, where $m \ll N$, which is comparable to vanilla GCN's complexity of $O(N^2)$. Specifically, we use the Nystrom method

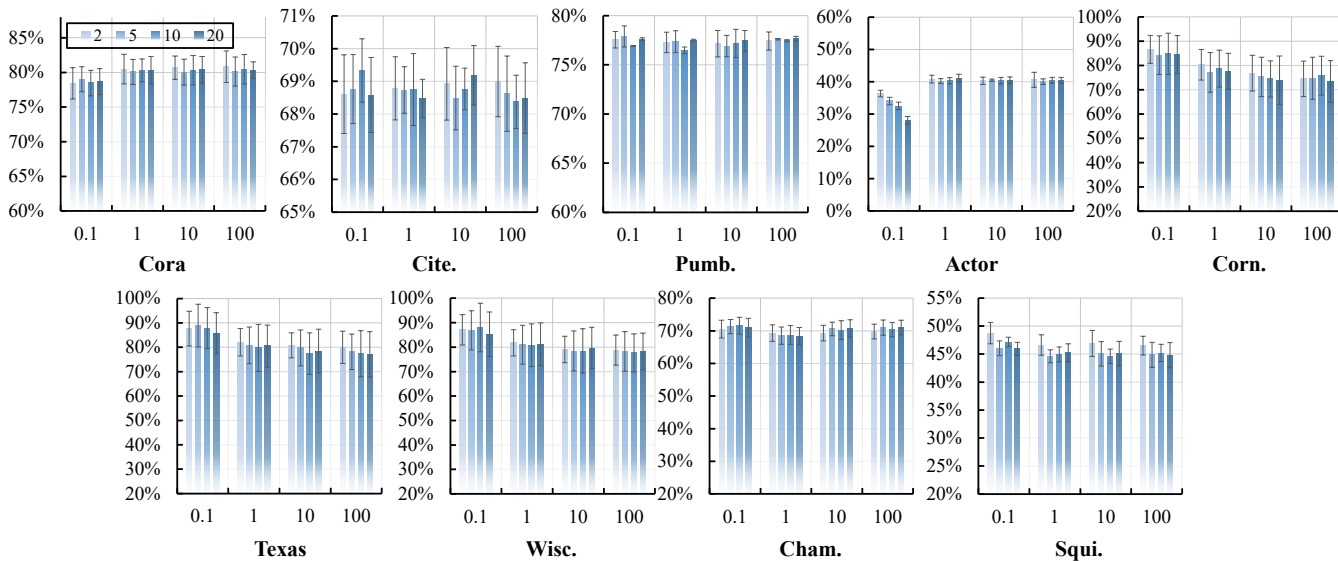

**Figure 7: Parameter analysis on $a_3$ on various number of layers ($\{2, 5, 10, 20\}$). Here $a_3$ is tuned in $\{0.1, 1, 10, 100\}$, and $a_2$ is fixed as 1. We report the average performance (and its standard deviation) of CSF across different numbers of layers, where the black line represents the standard variance.**

[27, 41] to approximate the inverse of the kernel and calculate the proposed high-pass spectral filter $K_{attr}$. Note that other kernel inverse approximation methods could also be applied. Given a kernel $K \in R^{N \times N}$, the Nystrom method first randomly samples $m \ll N$ columns to form a matrix $C \in R^{N \times m}$. Then, it builds a much smaller kernel matrix $Q \in R^{m \times m}$ based on the matrix $C$. As a result, the original kernel $K$ could be approximated by

$$K \approx C Q_k^{-1} C^T, \tag{6}$$

where $Q_k$ is the best rank-k approximation of $Q$, and $Q^{-1}$ is the (pseudo) inverse of $Q$. In our case, we approximate the inverse of $K$ by

$$K^{-1} \approx C_1^T Q_k C_1, \tag{7}$$

where $C_1$ is the pseudo inverse of $C$. The computational complexity for $K^{-1}$ is reduced from $O(N^3)$ to $O(mN^2)$, where $m \ll N$.

We test our method on the two largest assortative and dis-assortative graphs in our experiments (i.e., Pubmed and Actor). To push the limit of the Nystrom method for kernel inverse approximation, we set the sample size $m$ and rank-k to be 0.1% of the original data for simplicity. As shown in Table 5, we also report the computation time for calculating the cross-space filter, in addition to the model accuracy. The results show that the Nystrom-based CSF substantially increases calculation efficiency while maintaining some level of model effectiveness.

Lastly, we would like to emphasize that the kernel and inverse kernel have good characteristics and are widely used in various applications such as deblurring images [37] and interpreting deep neural networks [17]. By using the kernel method, we can interpret our high-pass filter and improve the effectiveness of GCN in addressing the over-smoothing problem.

**Table 5: CSF with Nystrom approximation. '-' means vanilla CSF. $m = M * \#Nodes$. For simplicity, the experiments are conducted with ten layers.**

| M | Pubm. (#Nodes=19717) | | Actor (#Node=7600) | |
|---|---|---|---|---|
| (%) | Acc. | Time (s) | Acc. | Time (s) |
| 0.1 | 76.53±1.46 | 2.25 | 48.84±1.32 | 1.76 |
| − | 76.72±1.41 | 5.16 | 49.34±1.48 | 1.99 |

## E APPENDIX FOR OVER-SMOOTHING PROBLEM EVALUATION

As discussed in Section 5, we design experiments to evaluate the performances of all methods on various data sets, together with the number of layers ranging from $\{2, 5, 10, 20\}$. The overall results are provided in Table 6 and the average version is provided in Table 1. Note that we omit the results of GAT and GCN-BC if they overflow the run-time memory. Basically, most methods suffer from decreased performance, while our method and some adaptive filter-based methods are more robust to the over-smoothing problem. As the number of model layers increases, the performance of the FAGCN and AKGNN methods is more stable than that of PGNN and AdaGNN. However, due to the lack of node attribute support, the overall performance of FAGCN and AKGNN is still not as good as CSF.

**Table 6: Over-smoothing problem evaluation of all methods. 'OOM' means 'out-of-memory'.**

| Model | #Layer | Cora | Cite. | Pubm. | Actor | Corn. | Texas | Wisc. | Cham. | Squi. |
|---|---|---|---|---|---|---|---|---|---|---|
| MLP | 2 | 53.01±1.70 | 51.98±2.48 | 66.71±1.70 | 45.76±0.76 | 83.42±6.45 | 84.21±7.94 | **85.29±6.62** | 61.21±2.09 | 49.74±1.66 |
| GCN | 2 | 80.03±2.02 | 67.79±1.22 | 75.01±1.88 | 33.99±1.03 | 62.37±8.78 | 69.74±8.16 | 65.88±8.63 | 62.46±2.32 | 34.97±2.32 |
| SGC | 2 | 79.59±1.63 | 67.95±0.86 | 75.06±2.06 | 34.47±0.83 | 65.00±9.20 | 70.26±9.77 | 67.06±8.31 | 63.93±2.09 | 37.66±2.01 |
| GAT | 2 | 63.68±1.61 | 58.65±2.37 | 74.84±2.01 | 30.91±0.98 | 63.68±8.49 | 67.37±7.15 | 60.20±8.57 | 60.86±2.01 | 37.66±1.77 |
| GraphSAGE | 2 | 45.51±3.91 | 42.34±6.30 | 60.12±5.02 | 30.18±0.83 | 59.21±8.44 | 61.58±10.47 | 57.84±9.88 | 54.96±2.97 | 33.10±2.89 |
| APPNP | 2 | 80.62±1.97 | 68.82±0.69 | 75.61±2.35 | 29.93±0.87 | 61.05±7.83 | 65.26±8.58 | 59.61±7.80 | 53.00±2.56 | 21.65±0.49 |
| JKNET | 2 | 77.43±1.69 | 65.84±1.73 | 74.92±1.94 | 28.18±2.16 | 60.79±7.59 | 63.95±8.33 | 61.57±7.46 | 56.58±2.72 | 33.32±1.37 |
| GCN-BC | 2 | 42.54±2.73 | 38.71±2.97 | 73.36±2.88 | 28.09±1.11 | 65.00±8.78 | 67.63±8.78 | 70.20±7.83 | 67.08±1.68 | 48.13±1.54 |
| FAGCN | 2 | 78.19±1.81 | 65.15±2.72 | 74.53±1.91 | 39.51±1.12 | 72.37±7.77 | 78.68±8.45 | 78.04±8.46 | 68.51±2.27 | 44.79±2.27 |
| PGNN | 2 | 72.15±2.96 | 47.55±2.36 | 64.56±2.28 | 39.74±0.90 | 61.05±6.42 | 65.26±7.73 | 57.06±6.37 | 60.81±1.58 | 44.08±1.99 |
| AdaGNN | 2 | 73.23±1.97 | 63.43±2.39 | 69.77±4.46 | **48.38±1.73** | 80.26±6.93 | 84.21±5.68 | 84.12±5.43 | **73.51±2.52** | **57.68±1.33** |
| AKGNN | 2 | 72.35±4.19 | 64.48±2.36 | 71.05±4.66 | 34.98±1.09 | 59.47±7.67 | 60.79±7.28 | 61.76±7.23 | 61.75±1.95 | 36.60±1.45 |
| CSF | 2 | **81.45±1.61** | **68.94±0.84** | **76.41±1.70** | 47.26±1.85 | **86.58±5.03** | **89.47±6.20** | 89.22±6.08 | 71.27±2.95 | 57.03±1.04 |
| CSF -w/o topology | 2 | 33.60±1.39 | 27.22±1.67 | 55.39±2.91 | 46.89±0.80 | 86.84±7.13 | 86.58±6.26 | 88.63±6.39 | 62.94±2.30 | 51.05±1.75 |
| CSF -w low-pass attribute | 2 | 81.43±1.63 | 68.84±0.81 | 77.40±1.86 | 33.61±0.65 | 63.95±8.42 | 70.00±7.67 | 64.71±8.42 | 58.82±1.75 | 36.12±1.07 |
| CSF -w only low-pass attribute | 2 | 31.33±0.79 | 21.41±1.08 | 50.18±0.83 | 28.25±0.77 | 60.53±9.12 | 60.53±8.95 | 57.84±8.78 | 28.31±3.07 | 20.46±2.31 |
| MLP | 5 | 34.22±1.93 | 28.45±3.10 | 56.37±1.91 | 41.00±1.65 | 71.05±8.41 | 71.05±7.85 | 72.75±5.73 | 41.78±5.11 | 27.29±6.68 |
| GCN | 5 | 52.78±5.52 | 50.43±4.50 | 67.68±3.74 | 25.70±1.24 | 58.68±8.14 | 58.68±8.14 | 48.04±4.74 | 32.61±10.88 | 20.12±1.10 |
| SGC | 5 | 78.44±1.65 | 62.70±2.44 | 74.62±1.76 | 28.18±1.26 | 60.53±8.59 | 62.63±8.30 | 54.71±7.13 | 52.70±2.65 | 30.02±2.51 |
| GAT | 5 | 64.11±1.91 | 58.83±3.99 | OOM | 30.67±1.14 | 62.63±9.18 | 66.58±7.95 | 61.37±8.72 | 60.83±2.38 | 37.40±1.91 |
| GraphSAGE | 5 | 45.04±2.84 | 41.49±6.62 | 61.07±3.90 | 30.17±0.64 | 59.21±8.34 | 61.84±7.67 | 56.67±8.64 | 55.29±2.98 | 32.13±2.75 |
| APPNP | 5 | 80.62±1.97 | 68.34±1.11 | 76.33±2.14 | 26.32±0.78 | 58.95±8.34 | 58.95±8.61 | 53.14±8.03 | 37.32±2.65 | 21.03±0.99 |
| JKNET | 5 | 78.00±2.36 | 65.98±1.53 | 74.93±1.95 | 26.34±1.12 | 58.95±8.25 | 60.26±7.99 | 57.45±5.39 | 46.73±4.00 | 30.67±2.02 |
| GCN-BC | 5 | 29.70±5.62 | 20.83±1.18 | OOM | 10.90±0.75 | 67.37±10.02 | 67.37±8.15 | 73.14±5.99 | 23.07±2.76 | 22.89±1.56 |
| FAGCN | 5 | 80.70±2.01 | 66.10±1.56 | 75.52±2.93 | 39.82±1.27 | 73.95±7.69 | 78.95±8.32 | 78.82±9.00 | 63.97±2.53 | 42.83±2.45 |
| PGNN | 5 | 71.05±3.10 | 48.80±2.21 | 67.89±2.31 | 26.45±0.98 | 60.00±8.02 | 64.21±7.67 | 56.86±6.67 | 43.05±2.92 | 32.71±1.59 |
| AdaGNN | 5 | 41.06±1.86 | 31.99±0.90 | 51.50±2.35 | 34.36±1.56 | 60.53±8.68 | 75.26±10.17 | 68.43±7.98 | 65.90±2.83 | 50.45±1.41 |
| AKGNN | 5 | 77.93±1.93 | 66.08±3.07 | 74.87±1.87 | 35.28±0.80 | 59.74±8.51 | 62.11±9.05 | 60.20±8.27 | 61.45±2.39 | 37.74±2.07 |
| CSF | 5 | 80.88±1.85 | 68.61±1.31 | 76.98±1.54 | 49.27±1.46 | 83.68±6.77 | 88.16±7.15 | 87.25±6.93 | 73.33±2.65 | 60.64±1.38 |
| CSF -w/o topology | 5 | 34.09±1.15 | 26.52±2.24 | 55.14±4.24 | 49.54±0.57 | 86.58±6.96 | 86.58±6.50 | 87.25±6.75 | 63.51±1.85 | 54.49±1.27 |
| CSF -w low-pass attribute | 5 | 80.76±1.64 | 68.28±1.00 | 77.64±1.21 | 34.23±0.82 | 62.37±8.78 | 68.68±8.18 | 63.73±8.93 | 56.78±1.84 | 34.74±0.97 |
| CSF -w only low-pass attribute | 5 | 31.37±0.82 | 21.43±1.24 | 49.88±0.77 | 28.28±0.75 | 60.53±9.45 | 58.95±7.96 | 57.65±10.83 | 29.34±2.29 | 22.83±1.48 |
| MLP | 10 | 31.30±0.76 | 20.83±0.78 | 39.88±0.25 | 25.70±1.24 | 58.68±8.14 | 58.68±8.14 | 46.47±6.06 | 21.51±0.84 | 20.35±1.05 |
| GCN | 10 | 31.30±0.76 | 21.53±2.02 | 40.49±1.92 | 25.41±1.37 | 58.68±8.14 | 58.68±8.14 | 46.47±6.06 | 21.45±0.89 | 20.17±1.04 |
| SGC | 10 | 65.42±4.71 | 48.22±4.01 | 72.76±2.50 | 25.92±1.37 | 59.47±8.25 | 60.53±8.59 | 51.18±6.57 | 35.11±3.33 | 22.91±1.15 |
| GAT | 10 | 64.07±1.86 | 58.62±2.99 | OOM | 30.32±0.79 | 63.95±10.23 | 66.84±9.70 | 59.61±7.63 | 60.99±1.96 | 37.97±2.16 |
| GraphSAGE | 10 | 44.89±2.99 | 41.09±6.57 | 59.96±4.32 | 30.14±0.66 | 59.74±8.04 | 62.37±8.04 | 57.06±7.98 | 55.24±2.47 | 32.49±2.07 |
| APPNP | 10 | 78.21±1.46 | 68.02±1.12 | 76.24±2.37 | 25.89±1.17 | 58.68±8.14 | 58.68±8.14 | 53.53±7.16 | 33.53±2.92 | 20.74±1.08 |
| JKNET | 10 | 76.49±2.23 | 65.44±1.77 | 75.04±2.18 | 26.22±1.25 | 58.95±7.67 | 60.00±8.93 | 57.25±5.61 | 37.19±2.35 | 29.07±1.38 |
| GCN-BC | 10 | 10.65±0.59 | 9.86±5.07 | OOM | 10.90±0.75 | 66.58±9.85 | 67.89±8.93 | 72.55±7.34 | 21.21±1.66 | 20.37±1.11 |
| FAGCN | 10 | **82.23±1.99** | 66.90±1.84 | 76.11±2.10 | 39.81±1.45 | 76.32±9.03 | 78.16±8.60 | 78.63±8.29 | 63.11±1.51 | 43.26±1.46 |
| PGNN | 10 | 59.86±4.75 | 45.79±2.81 | 67.27±3.15 | 26.41±0.90 | 58.95±8.61 | 60.00±8.40 | 54.31±5.85 | 34.52±4.49 | 30.31±1.73 |
| AdaGNN | 10 | 32.35±1.31 | 22.66±2.27 | 49.79±4.85 | 26.74±1.32 | 60.26±6.73 | 64.21±9.94 | 63.53±7.91 | 39.89±3.67 | 24.92±2.72 |
| AKGNN | 10 | 80.22±0.98 | 66.70±2.09 | 75.07±2.64 | 35.22±0.88 | 58.95±7.96 | 61.58±8.25 | 56.08±8.02 | 61.58±2.02 | 37.51±2.35 |
| CSF | 10 | 80.80±1.89 | 68.33±0.85 | 76.72±1.41 | 49.34±1.48 | 85.26±5.98 | 87.63±6.57 | 86.47±7.65 | 73.05±2.54 | 61.04±1.17 |
| CSF -w/o topology | 10 | 33.88±1.01 | 27.51±1.74 | 55.90±2.36 | 49.38±0.66 | 86.58±6.26 | 86.84±6.56 | 87.84±6.46 | 63.29±2.08 | 54.06±1.52 |
| CSF -w low-pass attribute | 10 | 80.61±1.95 | 68.32±1.07 | 77.41±1.07 | 34.32±0.92 | 63.16±8.50 | 70.79±8.55 | 63.73±8.88 | 57.87±2.47 | 34.84±1.19 |
| CSF -w only low-pass attribute | 10 | 31.27±0.82 | 21.12±0.89 | 51.10±0.89 | 28.18±0.78 | 59.21±8.62 | 59.47±9.30 | 62.75±9.96 | 29.93±2.50 | 21.07±2.53 |
| MLP | 20 | 31.30±0.76 | 20.89±0.73 | 39.88±0.25 | 25.70±1.24 | 58.68±8.14 | 58.68±8.14 | 46.47±6.06 | 21.51±0.84 | 20.26±1.04 |
| GCN | 20 | 31.30±0.76 | 20.83±0.78 | 39.88±0.25 | 25.41±1.37 | 58.68±8.14 | 58.68±8.14 | 46.47±6.06 | 21.51±0.84 | 20.32±1.03 |
| SGC | 20 | 38.77±2.77 | 32.18±2.60 | 61.98±4.92 | 25.46±1.34 | 59.74±7.55 | 58.68±8.14 | 47.65±7.28 | 31.67±2.72 | 21.82±0.99 |
| GAT | 20 | 64.16±2.60 | 59.41±8.27 | OOM | 30.41±1.12 | 63.42±7.79 | 65.79±8.23 | 59.41±8.27 | OOM | OOM |
| GraphSAGE | 20 | 45.59±3.05 | 42.96±5.76 | 60.21±5.22 | 30.26±0.77 | 59.47±8.43 | 61.32±9.53 | 56.08±7.35 | 55.07±2.85 | 32.19±2.64 |
| APPNP | 20 | 74.02±1.82 | 67.46±1.55 | 73.43±2.35 | 26.01±1.11 | 58.68±8.14 | 58.68±8.14 | 46.47±6.06 | 32.57±2.37 | 20.61±1.24 |
| JKNET | 20 | 73.84±2.25 | 64.63±2.23 | 74.30±2.07 | 26.24±0.93 | 58.68±8.14 | 59.74±8.69 | 50.78±6.63 | 35.24±2.95 | 28.19±1.35 |
| GCN-BC | 20 | 10.65±0.59 | 7.44±0.50 | OOM | 10.90±0.75 | 65.26±9.18 | 67.11±9.55 | 72.94±6.39 | 21.21±1.66 | 20.37±1.11 |
| FAGCN | 20 | **82.41±1.48** | 67.25±1.14 | 75.88±2.23 | 39.49±1.23 | 72.63±7.57 | 78.68±8.45 | 79.61±9.25 | 64.06±2.30 | 42.77±1.77 |
| PGNN | 20 | 37.80±6.88 | 29.77±4.92 | 45.48±7.22 | 26.45±1.01 | 58.95±7.67 | 60.26±9.24 | 52.16±5.33 | 28.49±2.70 | 29.98±1.09 |
| AdaGNN | 20 | 31.47±0.70 | 21.12±0.71 | 42.73±2.84 | 27.12±1.19 | 58.68±8.14 | 58.68±8.14 | 49.41±9.09 | 30.04±2.38 | 22.82±1.05 |
| AKGNN | 20 | 80.74±2.28 | 66.08±1.40 | 75.41±2.42 | 35.26±0.78 | 58.68±8.14 | 62.89±8.81 | 57.65±6.93 | 60.35±2.18 | 37.38±1.23 |
| CSF | 20 | 81.21±1.75 | 68.26±1.08 | 77.06±1.35 | 48.86±1.66 | 83.42±8.14 | 87.63±7.02 | 86.47±7.19 | 73.31±2.26 | 60.58±1.17 |
| CSF -w/o topology | 20 | 33.50±1.27 | 27.16±2.06 | 55.47±3.30 | 49.49±0.85 | 86.58±6.73 | 86.32±6.88 | 87.06±6.61 | 63.38±1.73 | 54.15±1.30 |
| CSF -w low-pass attribute | 20 | 80.68±1.82 | 68.07±1.31 | 77.52±2.07 | 34.17±0.75 | 63.95±7.85 | 68.68±8.81 | 63.73±7.96 | 57.57±2.42 | 34.76±1.50 |
| CSF -w only low-pass attribute | 20 | 31.27±0.82 | 21.15±0.67 | 51.56±0.57 | 28.19±0.64 | 60.26±8.36 | 60.53±8.86 | 67.84±7.96 | 29.98±2.52 | 23.44±1.76 |

