# OpenReview forum: "Cross-Space Adaptive Filter: Integrating Graph Topology and Node Attributes for Alleviating the Over-smoothing Problem"
_ACM.org/TheWebConf/2024/Conference — TheWebConf24_

### Official Review · Reviewer_dLdN · 2023-11-23

**Novelty:** 5
**Technical Quality:** 6

**Review:**

This paper studies the over-smoothing problem of graph convolutional networks. Specifically, the authors propose an interpretable high-pass filter to capture the correlations among node attributes and extract high-frequency information from the attribution space. By integrating the high-pass filter and the low-pass filter from graph topology space, the proposed cross-space adaptive filter (CSF) shows promising performance on addressing the over-smoothing problem, and helps GCN models achieve superior performance on the challenging disassortative graphs.

Pros:
1. This paper is well organized and easy to follow.

2. This paper explores the over-smoothing problem of GCNs --- an essential challenge in the filed of graph neural network, and propose an effective solution.

3. This work can be interpreted theoretically.

Cons:
1. The authors do not explain why the high-pass filter come from the attribute space, while low-pass filter come from the topology space. The motivation of such a design is desired to be clarified.

2. The authors conduct experiments on the disassortative graphs (i.e., heterophilic graphs). More related baselines such as [1] and [2] should be considered for comparison.


[1] Auto-HeG: Automated Graph Neural Network on Heterophilic Graphs. WWW 2023.

[2] Convolutional Neural Networks on Graphs with Chebyshev Approximation, Revisited. NeurIPS 2022.

**Questions:**

Please response according to the aforementioned cons.

**Ethics Review Description:**

No ethical issues

**Reviewer Confidence:**

3: The reviewer is confident but not certain that the evaluation is correct

**Scope:**

4: The work is relevant to the Web and to the track, and is of broad interest to the community

---

### Official Review · Reviewer_orgj · 2023-11-24

**Novelty:** 5
**Technical Quality:** 6

**Review:**

This paper aims to solve the over-smoothing and improve the effectiveness of deep GCN on downstream tasks. It designs a cross-space filter named CSF. Specifically, it first designs a high-pass filter based on correlations of node attributes, which can be interpreted as a minimization of semi-supervised kernel ridge regression. It then combines the high-pass filter with the conventional low-pass filter of GNNs. Finally, an effective multiple-kernel learning strategy is employed to unify these two filters and adaptively control the frequency information. Extensive experiments demonstrate the effectiveness of CSF.

> Quality

Pros: Most arguments are well-supported. Extensive experiments demonstrate the effectiveness of the proposed method. This paper displays a variety of visualization reports, e.g., Figures 2 and 3, which make the model more accessible and convincing.

Cons: This paper doesn’t compare with AM-GCN [1], which similarly unifies the graph topology and node attributes. In addition, this paper lacks some important baselines and an analysis of $\gamma$ in Eq(5). Please see Questions 2.

> Clarity

Pros: The paper is well-organized in general and experimental details are clear.

Cons: The paper lacks a theoretical derivation process of the equations in Session 4.1. Additionally, there are some minor mistakes in these equations. Please see Question 1.

> Originality

This paper solves the over-smoothing problem via a cross-space filter, which adaptively integrates an attributed-based high-pass filter and a topology-based low-pass filter to capture the frequency information.

> Significance

The lack of interpretability of hand-crafted high-pass filters is a significant challenge. This paper is the first to interpret the attribute-based high-pass as a minimization of semi-supervised kernel ridge regression.

[1] AM-GCN: Adaptive multi-channel graph convolutional networks. KDD 2020.

**Questions:**

1. There are some minor mistakes in equations 2 and 3. Specifically, the first term should be $||\mathbb{Y}^T\left(I-\Gamma\left(K, a_3\right)\right)\mathbb{Y}||_{F}$ rather than $\mathbb{Y}^T\left(I-\Gamma\left(K, a_3\right)\right)\mathbb{Y}$ if $\mathbb{Y}$ is a matrix.
Additionally, I think it would be better to provide the derivation process of the equations in Session 4.1, especially the process from Eq 1 to Eq 2 and the solution for Eq 3.

2. The benchmarks lack experiments evaluating polynomial-based spectral GNNs, such as GPR-GNN [1] and ChebNetII [2], as well as GNNs with cross-space convolution, such as AM-GCN. GPR-GNN and ChebNetII not only solve the over-smoothing problems but also demonstrate competitive performance on the disassortative graphs.

[1] Adaptive universal generalized pagerank graph neural network.

[2] Convolutional neural networks on graphs with Chebyshev approximation, revisited.

**Ethics Review Description:**

No ethics issue

**Reviewer Confidence:**

3: The reviewer is confident but not certain that the evaluation is correct

**Scope:**

3: The work is somewhat relevant to the Web and to the track, and is of narrow interest to a sub-community

---

### Official Review · Reviewer_SCn8 · 2023-11-26

**Novelty:** 4
**Technical Quality:** 4

**Review:**

Based on the multiple-kernel learning, this paper proposes a cross-space adaptive filter that includes attributed-based high-pass filter and topology-based low-pass filter, which can alleviate the over-smoothing problem, especially when processing disassortative graphs. Besides, the proposed attributed-based filter is interpretable based on semi-supervised kernel ridge regression. Experiments demonstrate that the proposed CSF can alleviate the over-smoothing problem as GNN goes deep.

Strength:

S1. This paper attempts to alleviate the over-smoothing problem of GNNs from the perspectives of graph kernel theory, where both attribute information and topology structure are employed to design different filters. This idea is sound and interesting.

S2.  This paper provides theoretical analysis for the proposed attributed-based high-pass filter, which is convincing.

S3. The experimental results compared to baselines are promising.

Weakness:

W1. In a semi-supervised learning setting, how to process unlabeled nodes in Z in Eq (1) in this paper? would their labels be inferred during model training? If not, how does this approach differ from supervised Kernel Ridge Regression?

W2. GNN-BC [1] has validated that attribute information and topological structure can be simultaneously employed to alleviate the over-smoothing issue in GNNs. This paper should clarify the advantages of employing both information in graph kernel theory compared to GNN-BC.

Furthermore, the performance of GNN-BC reported in this manuscript is considerably lower than that in the original paper. While the authors mention the absence of available code for GNN-BC, its code was made available at [2] last year. Please check.

W3.	The paper shows that each filter is important for graph learning. However, in this cross-space filter, it should be clarified how to achieve a balance between the high-pass and low-pass filters. Furthermore, an experimental analysis of the CSF performance across various adaptive scenarios would be insightful.

Reference:
[1] Liang Yang, Wenmiao Zhou, Weihang Peng, Bingxin Niu, Junhua Gu, Chuan Wang, Xiaochun Cao, Dongxiao He. "Graph Neural Networks Beyond Compromise Between Attribute and Topology". WWW 2022.
[2] https://github.com/GitEventHandler/GNNBC

**Questions:**

Please refer to weakness above.

**Reviewer Confidence:**

3: The reviewer is confident but not certain that the evaluation is correct

**Scope:**

3: The work is somewhat relevant to the Web and to the track, and is of narrow interest to a sub-community

---

### Official Review · Reviewer_tnra · 2023-11-30

**Novelty:** 4
**Technical Quality:** 4

**Review:**

The paper introduces a novel method, Cross-Space Adaptive Filter (CSF), designed to enhance the performance of deepGCNs in node classification tasks. The authors propose a attribute-based high-pass filter and a topology-based low-pass filter, combining them to capture adaptive-frequency information. The construction of the proposed high-pass filter is interpreted through kernel ridge regression, offering a new approach to alleviate the over-smoothing problem. The authors conduct comparative experiments with various baselines under different
numbers of convolution layers and reveal more characteristics of CSF by conducting ablation studies on graph topology and node attributes. The experimental results demonstrate CSF's ability to mitigate the over-smoothing issue and enhance the effectiveness of deep GCN.
However, the writing can be improved, especially in section 1 and 2, there some almost identical sentences were repeated many times. In summary, the paper exhibits good novelty, solid analysis, and readability.

**Questions:**

In the process of node updating, there is a concatenation with the raw feature X. Can the authors provide an ablation study by comparing the performance with a GCN that includes such a skip connection?

**Reviewer Confidence:**

2: The reviewer is willing to defend the evaluation, but it is likely that the reviewer did not understand parts of the paper

**Scope:**

3: The work is somewhat relevant to the Web and to the track, and is of narrow interest to a sub-community

---

### Official Review · Reviewer_Nnvd · 2023-12-01

**Novelty:** 4
**Technical Quality:** 4

**Review:**

This paper proposes to a novel method for alleviating the over-smoothing problem in GCN by applying a low-pass filter in the graph topology space and a high-pass filter in the node attribute space.

pros:
1. The paper is well-written and easy to follow.
2. The method is supported with comprehensive theory.
3. The experiments demonstrate good improvements over baselines.

cons:
1. The method has high time complexity of $O(n^3)$, which makes it infeasible for medium and large graphs.
2. The experimental setup seems biased towards the proposed method. Table 1 shows the average classification accuracy over models with varying depth. However, it is well-known that most GNN architectures achieve the best accuracy at 2 or 3 layers and experience serious accuracy degradation with increasing depth. As shown in figure 4, the proposed method CSF does not achieve better accuracy with increasing depth in most cases. So having many layers seems very unnecessary.
3. The authors are missing on some important baselines, e.g. RevGNN [1], which also uses a deep architecture.

References

[1] Li, Guohao, et al. "Training graph neural networks with 1000 layers." International conference on machine learning. PMLR, 2021.

**Questions:**

Please see the full review for questions.

**Reviewer Confidence:**

2: The reviewer is willing to defend the evaluation, but it is likely that the reviewer did not understand parts of the paper

**Scope:**

3: The work is somewhat relevant to the Web and to the track, and is of narrow interest to a sub-community

---

### Decision · Program_Chairs · 2024-01-22

**Decision:**

Accept

**Comment:**

The paper develops a method to tackle the over-smoothing problem in graph learning. Specifically, the method uses an attribute-based high-pass filter and a topology-based low-pass filter.

 Pros:
 * Oversmoothing is a well-known problem in graph learning and the method developed in the paper shows pretty good performance over the baseline especially when the number of layers increases.
 * The paper is easy to follow and understand.
 * The method is novel and has theoretical support.

 Cons:
 * The method is only studied on small datasets, but the real-world datasets in a single graph setting are usually very large. Even though the author claims that the method has a complexity of O(m x N x N), this complexity is too large to scale to large datasets. This limits the applicability of the method in the real-world setting.
 * The method doesn't really increase the model performance when the number of layers increases. The authors should find the settings/datasets where increasing the number of layers can improve performance. Otherwise, the studied method is not useful in practice.